# The Preparation Process, Microstructure and Properties of Cellular TiC-High Mn Steel-Bonded Carbide

**DOI:** 10.3390/ma13030757

**Published:** 2020-02-07

**Authors:** Guoping Li, Haojun Zhou, Hao Yang, Mingchu Huang, Yingbiao Peng, Fenghua Luo

**Affiliations:** 1State Key Laboratory of Powder Metallurgy, Central South University, Changsha 410083, China; guopingli@csu.edu.cn (G.L.); zhouhaojunlife@163.com (H.Z.); yanghaoicsy@163.com (H.Y.); h13657483787@163.com (M.H.); 2Laiwu Vocational and Technical College, Jinan 271100, Shandong, China; 3College of Metallurgical and Materials Engineering, Hunan University of Technology, Zhuzhou 412008, Hunan, China

**Keywords:** TiC, steel-bonded carbide, cellular structure, high Mn steel binder

## Abstract

TiC-high Mn steel-bonded carbide with a cellular structure was designed and fabricated by powder metallurgy techniques using coarse and fine TiC particles as the hard phase. This preparation process of the alloy was designed carefully and optimized. The microstructure of the alloy was observed using a scanning electron microscope. The results show that there are two types of microstructures observed in this TiC steel-bonded carbide: the coarse-grained TiC structure and fine-grained TiC structure. The transverse rupture strength and impact toughness of the alloy reach maximum values 2231 MPa and 12.87 J/cm^2^, respectively, when the starting weight ratio of MP-A (containing coarse TiC particles) to MP-B (containing fine TiC particles) is 60:40. Hence, this study serves as a feasible and economical example to prepare a high-strength and high-toughness TiC-high Mn steel-bonded carbide with little production cost increase.

## 1. Introduction

Steel-bonded carbide is a new engineering material prepared according to conventional powder metallurgical techniques using WC and/or TiC as the hard phase and steel as the metallic binder [1,2]. Due to combining high hardness, good wear resistance, excellent oxidation resistant, superior chemical stability of the hard phase, and high toughness and suitable strength of the metallic binder, steel-bonded carbide is used in multiple applications. These applications include wear-resistant parts, extrusion dies and punches, high-speed milling, surface finishing operations, forming tools, and carbon and stainless steel machining [3,4,5,6,7]. Among all the TiC steel-bonded carbides, TiC-high Mn steel-bonded carbide is one of the most successful applications in oil production, mine exploration, coal mining and cement production [8,9]. TiC-high Mn steel-bonded carbide and high Mn steel matrix are cast into a whole composite part. Additionally, the chemical composition of the high Mn steel matrix is the same as the metallic binder of TiC-high Mn steel-bonded carbide. After heat treatment, the microstructure of binder phase in TiC steel-bonded carbide changes into austenite which exhibits a good ductility and toughness. Due to the work-hardening effect of high Mn steel, the binder microstructure of TiC steel-bonded carbide changes into martensite when the composite part is impacted strongly, making it hard and wear-resistant. This phase transformation of the high Mn steel matrix and the TiC steel-bonded carbide is synchronous in improving wear-resistance and prolonging the working life of the composite casting. Therefore, this composite casting is widely applicable in situations with strong impact and vibration. 

A prominent disadvantage of TiC- and/or TiCN-based cermets, including TiC steel-bonded carbide, is their brittleness due to the poor wettability of the metallic binder on the hard phase. This leads to the reduced toughness and moderate strength of TiC steel-bonded carbide compared to tungsten cemented carbide and WC steel-bonded carbide, thereby limiting its applications. Hence, it is crucial to improve the wettability between the binder on the hard phase in order to increase the phase interface bonding strength of the cermets. Studies have confirmed that Mo [10,11,12,13,14], Mo_2_C [14,15,16,17,18,19], WC [15,18,20,21,22,23], TaC [15,17,19], NbC [19], and ZrC [24] can improve the wettability of the metallic binderon the hard phase, refine the hard phase, and modify the mechanical properties of the cermets.

Despite implementing these conditions, the strength and toughness of TiC-based cermets should be further improved to widen their applications. Particularly, since TiC-high Mn steel-bonded carbide is mainly applied in strong impact working conditions, its strength and toughness should be further increased. Previously published results from the author have shown that using Fe-Mo-Cr, Fe-Mo pre-alloyed powders as the binders significantly improved the transverse rupture strength and impact toughness of TiC-high Mn steel-bonded carbide [4,25]. In the fields of geology, rock drilling and other wear- and impact-resistant conditions, it has confirmed that a dual composite WC-Co and a hybrid cemented carbide composite were applied which show excellent toughness under severe shock conditions [26,27] and these previous reports inspired this study. Moreover, research results from the author have shown that using Fe-Mo-Cr, Fe-Mo pre-alloyed powders as the binder provides more variable and selectable parameters to control and modify the mechanical properties of TiC-based cermets [4,25]; the variable and selectable parameters include the alloying elements and its amount, particle size, et al of the pre-alloyed powders which can offset some inconveniences and uncertainties during the preparation of adual composite WC-Co and a hybrid cemented carbide composite. Thus, it can be seen that none of the literature relates to information on the preparation, microstructure and properties of a cellular TiC-high Mn steel-bonded carbide. Hence, this study attempts to fill this research gap.

## 2. Research Method and Preparation of Materials

Two types of TiC particles, namely coarse and fine TiC particles, were chosen as ceramic particles for this study, and the TiC fisher sizes were 3.1–3.3 μm and 0.8–1.5 μm, respectively. The main characteristics of the raw powders are listed in Table 1. Figure 1 shows the SEM morphology of the two types of TiC particles.

The main purpose of this study was to obtain a type of inhomogeneous TiC-high Mn steel-bonded carbide deliberately. Within its microstructure, there is an aggregation area of fine-grained TiC, i.e., cellular microstructure. Therefore, in the process of alloy design, mixing and sintering, effective measures and steps should be taken to prevent the coarsening of fine TiC particles.

Nickel, ferromanganese, graphitic carbon and pre-alloyed iron powders are used as bonding phases. The pre-alloyed iron powders, prepared by water atomization method, are Fe-1.5Mo (200 mesh, ≤74 μm) and Fe-4.5Mo-3.75Cr-0.7C (100 mesh, ≤154 μm), respectively. Among them, Fe-1.5Mo pre-alloyed iron powder is used as the bonding phase of coarse TiC powder, while Fe-4.5Mo-3.75Cr-0.7C pre-alloyed iron powder is used as the bonding phase of fine TiC powder. Figure 2 shows the SEM morphology of the pre-alloyed iron powders, Figure 2a refers to the Fe-1.5Mo pre-alloyed powder, and its particle size is 25~67 μm; while Figure 2b refers to the Fe-4.5Mo-3.75Cr-0.7C pre-alloyed iron powder, and its particle size is 100~210 μm.

The raw powders are weighed and mixed to form two kinds of mixed powder. The composition and proportions of ingredients of the two kinds of mixed powder are listed in Table 2. The mixed powder containing coarse TiC particles was marked as MP-A (mixed powder A), while the mixed powder containing fine TiC particles was marked as MP-B (mixed powder B).

The design of MP-A is to form the coarse-grained TiC area. Hence, fine-grained Fe-1.5Mo pre-alloy powder is selected as the bonding phase to activate sintering and a lower grinding ball to powder weight ratio selected to avoid crushing and refining TiC particles excessively. MP-B is designed to form the fine-grained TiC area, so coarse-grained Fe-4.5Mo-3.75Cr-0.7C pre-alloy powder is selected as the bonding phase to repress the growth of TiC grains and a higher grinding ball to powder weight ratio selected to enhance the milling of TiC particles as finely as possible. Additionally, Cr element is added in the MP-B system because Cr can reduce the eutectic temperature of the alloy and influence the particle shape, particle size of ceramic phase in conjunction with Mo and/or C to modify the properties of TiC steel-bonded carbide [4].

After weighing their ingredients, MP-A and MP-B were mixed in a V-type mixer (Wuxi Xinbang Manufacturing of Powder Equipment Co., Ltd., Wuxi, China) for 120 min and then ball-milled in a planetary ball mill (Changsha Tianchuang Powder Technology Co., Ltd., Changsha, China) bathed in ethanol for 24 h at a rotation speed of 220 rpm. A stainless steel ball was selected as the grinder. To avoid excessive grind of coarse TiC particles, the grinding ball to powder weight ratio was 3:1 for MP-Awhile that of MP-B was 6:1 to mill fine TiC particles as much as possible. The milling time was similar for both materials for operational convenience and to guarantee uniformity of MP-A and MP-B, respectively.

The above wet grinding slurry was dried in a vacuum oven (Shanghai Dengsheng Instrument Manufacturing Co., Ltd., Shanghai, China) at 70 °C for 8 h when the milling processes were completed. Subsequently, 4 wt.% styrene-butadiene rubber (SBR) (Shouguang city fat special petroleum products co. Ltd., Weifang, China) was added in both mixed powders as a forming agent. The rubber was dissolved in gasoline to form a rubber-gasoline solution with a concentration of 10.83%, which can guarantee a uniform distribution of rubber in the powders. Both powders, MP-A and MP-B, were dried in a vacuum oven at 80 °C for 12 h to remove the gasoline and then were pressed into columnar green compacts (Φ20 × 20 mm^3^) at a pressure of 30 MPa. The green compacts were crushed with a crushing screen to get 60 mesh (≤250 μm) pre-granulated mixed powder. This process was named the block granulation method. In this process, the amount of rubber additive was higher than that of conventional TiC steel-bonded carbide, so the strength of the pre-granulated mixed powder was increased to prevent disintegration in the subsequent remixing process, which facilitated the formation of a honeycomb structure in the alloy.

The two types of pre-granulated mixed powders, MP-A and MP-B, were remixed according to the proportion listed in Table 3 to prepare four experimental samples.

MP-A and MP-B were weighed according to the proportion of Table 3 and remixed in a V-type mixer for 120 min, then 2 wt.% rubber (SBR) was added in the powder mixture. The adding mode and operation are the same as the former process and the gasoline added was removed in a vacuum oven at 80 °C for 12 h. The dried powder mixture was pressed into columnar compacts (Φ20 × 20 mm^3^) at a pressure of 15 MPa and then crushed with a crushing screen to obtain 20 mesh (≤830 μm) re-granulated mixed powder. This re-granulation operation is aimed at ensuring the uniform distribution of MP-A and MP-B in the re-granulated particles. The re-granulated powder was pressed into the columnar green compacts (Φ20 × 60 mm^3^) under a uniaxial pressure of 200 MPa. Subsequently, the columnar green compacts were vacuum-sintered at 1420 °C for 60 min to obtain the bulk samples.

Figure 3 represents the integrated dewaxing and sintering curve of the TiC-high Mn steel bonded carbide in the study.

As shown in Figure 3, in the dewaxing stage (250–650 °C), multiple insulation stages were set up to dewax the forming agent completely for its perfect viscidity and increased content. Herein, two important insulation stages were observed, the first at 1150 °C where the carbon reduced the inevitable small amount of metallic oxides in the Fe powder, and the second at 1300 °C, where the metallic binder became molten or semi-molten. In the latter stage of high-temperature insulation, the flow of liquid adhesive along the ceramic particle space is promoted in order to effectively separate the ceramic particles from each other and thus, facilitate the formation of a TiC “skeleton”. These sintering processes are designed especially for TiC-based cermets using Fe-Mo and Fe-Mo-Cr pre alloyed powders as the binders [4].

The preparation processes of the cellular TiC-high Mn steel-bonded carbide are summarized in Table 4.

Finally, the bulk specimens were machined by wire-electrode cutting for the test samples. The geometrical size of the test samples for hardness and SEM microstructure observation is 10 mm × 10 mm × 15 mm, and the cross section of 10 mm × 10 mm for test is prepared with the standards of metallographic observation. The Rockwell hardness of the sintered cellular TiC-high Mn steel-bonded carbide was measured using an HR-150B Rockwell hardness tester (Laizhou Huayin Instruments Co., Ltd., Yantai, China) under a load of 60 kg. The transverse rupture strength (TRS) of the alloy was measured by a WDW-100E universal material testing machine (Jinan Test Machine Co., Ltd., Jinan, China) with a specimen size of 5 mm × 5 mm × 35 mm, a span distance of 20 mm, and a cross head velocity of 0.5 mm/min. The impact toughness (IM) test was performed using a JBW impact testing machine (Jinan Shijin Group Ltd., Jinan, China) with a specimen size of 10 mm × 10 mm × 55 mm. The density of the bulk specimens was measured using the Archimedes method and composition analysis was carried out using energy dispersive spectroscopy (EDS). The fracture surface morphology was observed using a SEM in secondary electron (SE) mode. The tunneling electron microscopy lamellae were prepared from the pins using a focused-ion beam (FIB, Fei Czech Republic S.R.O., Hillsboro, OR, USA). A transmission electron microscope (TEM, Fei Czech Republic S.R.O., Hillsboro, OR, USA) was used to investigate the selected area’s electron diffraction and microstructural characteristics of particular areas of the cermets.

## 3. Results

Figure 4 shows the SEM images of the experimental samples. These images show that the microstructures of the samples containing MP-A and MP-B are not well distributed. The area marked with the red circles in Figure 4 represents the zone where a concentration of fine TiC particles was observed. Other than the fine TiC particles concentration zone, coarse TiC particles are the main components of the samples. The special structure of the fine TiC concentration zone is located in coarse TiC particles matrix which looks like a honeycomb, also called a cellular structure.

In order to confirm the chemical composition of fine-grained TiC zone and coarse-grained TiC zone, an EDS analysis was conducted. The testing position for fine-grained TiC zone and coarse-grained TiC zone was marked with a red “+” symbol and a green “+” symbol, respectively, as shown in Figure 4b (sample #2). Table 5 shows the EDS analysis results of the fine-grained TiC zone and the coarse-grained TiC zone.

Table 5 shows that Cr content in the fine-grained TiC zone is much higher than that of the coarse-grained TiC zone. Thus, this means that the fine-grained TiC zone is formed by MP-B, while the coarse-grained TiC zone is formed by MP-A, which is consistent with our design and expectation.

Table 5 also indicates that a deviation exists between the measured and designed value of elements Mo, Cr, and Ni. The measured Mo content in fine-grained TiC zone is much higher than that of coarse-grained TiC zone, which indicates that pre-alloying of the Mo element is quite necessary, for example, to form Fe-Mo or Fe-Mo-Cr pre-alloyed powders initially in order to make full use of the improvement effect of Mo element on the cermets.

The content deviation of Cr and Ni is attributed to the stainless balls used as the grinders of the planet ball mill because the worn fragments of the grinder are left in the alloy, thus increasing the content of Cr and Ni. Another possible reason for the deviation is the liquid binder of MP-B, which occurred in advance because of the lower eutectic temperature caused by Cr and C, seeping into the region of MP-A. The measurement method and instrument accuracy may have also contributed to these discrepancies.

The density of the sintered bulk samples was measured using the Archimedes method, and the relative density is the ratio of actual density to theoretical density, as shown in Table 6. It is noteworthy that the relative density of the samples decreased slightly with the increase of MP-B. One reason is that the surface adsorbed oxygen by fine TiC particle increases with increase in the MP-B. The second probable reason is that Cr content increases with increase in the MP-B and reduces the sinterability of the alloy because of the strong affinity of Cr with oxygen.

Table 7 lists the mechanical properties of TiC-high Mn steel-bonded carbide, including hardness, transverse rupture strength (TRS) and impact toughness (IM).

Table 7 shows that the hardness of the samples increases initially and then decreases with the growth of the cellular structure. Meanwhile, the hardness of sample #2 reached an ideal maximum value 64.3 HRC. The transverse rupture strength (TRS) and impact toughness (IM) of the #2 alloy also reached the maximum values of 2231 MPa and 12.87 J/cm^2^, respectively.

## 4. Discussion

The key process for forming the cellular TiC-high Mn steel-bonded carbide is aimed at preserving the primary characteristics of fine TiC particles. In addition to sintering temperature, which is the most important parameter and should thus be selected carefully, many other parameters were also selected carefully to ensure the formation of cellular structures. Fe-1.5Mo pre-alloyed powder is used as the binder for MP-A to guarantee uniform growth of the coarse TiC particles and to offset its excessive growth at a high sintering temperature. Pure Mo powder is also added to achieve the total designed Mo content of the alloy, which is expected to promote the growth of coarse TiC particles given that it is more active than that of Fe-Mo pre-alloyed powder. The final TiC particle size produced by MP-A depends on the initial TiC particle size, binder composition and characteristics, ball-milling technology, and sintering technology among others. TiC grain growth is the competitive result of these comprehensive influencing factors.

For MP-B, Fe-4.5Mo-3.75Cr-0.7C pre-alloyed powder is used as the binder to repress grain growth with increasing Mo content. The alloying elements Mo-Cr and Mo-Cr-C have a complex and profound effect on the TiC grain growth. Cr and C decrease the eutectic temperature of the alloy to promote TiC grain growth. However, Mo and Cr influence and control the particle size and shape of the TiC grains complexly [4]. Hence, like MP-A, the TiC grain growth behavior of MP-B is also dependent on the comprehensive influencing factors. After this precise design, MP-A and MP-B were mixed and sintered at the same temperature to achieve the different expected goals.

In the re-granulating operation, some granulated Cr-containing MP-B particles may be broken inevitably. The broken small MP-B particles are mixed with or attached to the MP-A granulated particles. The broken small MP-B particles grow to form small size cellular structures; perhaps other smaller MP-B particles disappear and integrate into MP-A completely. Therefore, a small amount of Cr was found in coarse-grained TiC zone, as shown in Table 6, which strengthens the binder and is beneficial to the properties of the designed cermets.

The areas of cellular structure were calculated using the Image 88 system in Figure 4. The areas of fine-grained TiC zones of the four experimental samples were 16.3%, 25.4%, 34.8% and 42.1%, respectively. Although the area of the fine-grained TiC zones increases with the addition of MP-B, the increased amount is not equal to the addition of MP-B granulated powder. It is indicated that the granulated MP-B particles were broken in the subsequent remixing, re-granulating, and pressing process. The broken small MP-B particles were integrated with coarse TiC particles, thus becoming a part of the coarse-grained TiC zone. Some regions with a small quantity of fine TiC particles are observed in Figure 4, which is proof of the broken MP-B particles. Additionally, the cellular structure size is evidently not completely consistent with the granulated particle size (60 mesh, 250 μm). A few cellular structures are larger than the granulated particle and the cellular structure shape is not round or elliptical (Figure 4a,d). Some cellular structures are significantly smaller than the granulated particles and the cellular structure shape is also not round or elliptical (Figure 4b,c). It also shows that it is difficult to preserve the granulated particle size even after adding excessive forming agent in the block granulation preparation process. How to control the cellular structure size, shape, number and its distribution is still a significant problem. However, in any case, this study is a very useful attempt at offering a solution.

The formation of the cellular structure of the alloy is definitely attributed to the initial fine TiC particles, strengthening ball-milling process, binder composition, and so on. Among these factors, binder composition plays an important role in the formation of the cellular structure, because the alloying elements, Cr, Mo, and C have a significant and complex effect on the sintering process. Cr is the most important element in the formation of the cellular structure as it reduces the eutectic temperature of the alloy and promote the appearance of liquid metallic binder in the initial stage of sintering. Its effect is more obvious when it is combined with the existence of carbon in Fe-Mo-Cr-C pre-alloyed powder. With the improving wettability effect of Mo on the ceramic phase in Fe-Mo-Cr-C pre-alloyed powder, the liquid binder can flow throughout the TiC particles space and effectively separate them from each other during the sintering process. The coalescence grain growth of TiC particles is primarily inhibited and the dissolution-precipitation grain growth is also repressed. Hence, the fine-grained TiC zones form eventually. On the other hand, Cr is a strong carbide forming element in the formation of Cr_3_C_2_ or compound carbide, and Cr_3_C_2_ is an effective grain growth inhibitor for WC cemented carbide, TiC- and/or TiCN-based cermets. While no Cr_3_C_2_ grain growth inhibitor forms in MP-A powder, the broken MP-B particles provide insufficient Cr to form Cr_3_C_2_ inhibitor, so the MP-A particles grow a coarse-grained TiC ceramic matrix, merging broken small MP-B particles. In addition, the design of MP-A particles and the preparation process is meant to promote the grain growth of the coarse-grained TiC zone.

However, due to the imperfect preparation process adopted in the study, it is difficult to obtain the inhomogeneous alloy such that the cellular structure distributes among the coarse-grained TiC zones uniformly. Here, the meaning of the inhomogeneous alloy should conclude reasonable number, proper size, perfect shape and good distribution of the cellular structure that can improve the mechanical properties of the cermets. Although the microstructures in Figure 4b,c look more uniform than those in Figure 4a,d, they cannot be representative of the overall and actual microstructure of these cermets due to the limited microstructure observation ranges. With the increase of MP-B particles, the uncertainty in the inhomogeneity of the microstructure of the alloys increases, which has negative effects on the mechanical properties. In addition, the change in density may cause the loss of strength and hardness.

Figure 5 shows the impact fracture morphology of Samples #1 and #2 observed by SEM. It shows that trans-crystalline rupture is the dominant mode within the coarse-grained TiC zone and cleavage fracture is observed through some coarse TiC particles, marked with the green symbol “☐”. Trans-granular and inter-granular fractures are found in the fine-grained TiC zones, marked with the red symbol, “О”. The binder ductile fracture of the fine-grained TiC zone is a significant contributor to the toughness of the alloy in the crack-penetrating stage [28]. In the fracture stage, the trans-crystalline rupture (coarse and fine TiC particles) of the alloy was the dominant mechanism in improving the toughness because of the high rigidity of ceramic phase [29]. However, the area with fine-grained TiC broken zones was larger than that with coarse-grained TiC zones, which confirms that the cellular alloy structure is more important to the impact toughness.

Some effective attempts at improving toughness for TiC- and/or TiCN-based cermets are the character modification of the ceramic phase. The toughness of the cermets depends not only on the binder but also on the ceramic phase. It is clear that the hard phase plays a more important role in improving the toughness because trans-granular fracture is the dominate mode of TiC steel-bonded carbide. Many studies have shown that the properties of TiC- and TiCN-based cermets with WC additive improved effectively because WC improves the wettability of the binder on the ceramic phase, refines the ceramic phase, and strengthens the binder [15,22,30]. On the other hand, the TiC hard phase and added WC react during sintering to form (Ti,W)C rim structure on the surface of TiC particles and the rigidity of the hard phase increases because of the high rigidity of WC. Accordingly, the impact toughness of the TiC- and/or TiCN-based cermets increases due to WC addition if the thickness of the (Ti,W)C rim structure is appropriate. The transverse rupture strength and impact toughness of TiC- and/or TiCN-based cermets (TiC steel-bonded carbide included) are inferior to those of the conventional WC cemented carbide or WC steel-bonded carbide. One reason for this is the poor wettability of the binder on the ceramic phase. The intrinsic disadvantage of the low rigidity of TiC is also a possible reason for this inferiority.

Considering the practical application of the alloy, the ceramic particle size of the alloy should be increased adequately because big ceramic particles withstand stronger impact loads better than small ceramic particles. In practical application, a cellular structure is regarded as one big particle to undergo the external applied load. When the cracks occur and penetrate through a cellular structure, they penetrate preferably through the binder because of the fine TiC particles which consumes more energy to improves the toughness of the alloy. When the cracks spread into the coarse-grained TiC zone and encounter TiC particles, these large particles force the cracks to change the spreading direction and likely penetrate through the binder continually. Therefore, the binder of the alloy plays a dominant role in toughness improvement at the crack penetrating stage. However, at the fracture stage, the hard phase characteristics, particle shape and size of the alloy play the dominant roles. The toughness of the alloy is greatly improved because more TiC particles are split due to the bigger particle size and angular particle shape, as the rigidity of TiC is much higher than that of the binder, which consumes more energy during fracture.

In order to further investigate the strengthening mechanism on the TRS and IM improvement of the alloy, TEM analysis was carried out to examine whether the new phase formed in the binder because of Fe-Mo and Fe-Mo-Cr-C pre-alloyed powders were used as the binder.

Figure 6b shows the TEM electron diffraction pattern analysis of the metallic binder of the coarse-grained TiC zone. The results show that the bonding phase was a single austenite structure, no other compounds were found, and it had a favorable metallurgical combination with TiC particles. Figure 6c,d shows the TEM bright image and electron diffraction pattern analysis of the metallic binder of the fine-grained TiC zone. The results show that the TiC particle size is about 1 μm, which is similar to the particle size of the raw materials, indicating that TiC particles undergo no significant change during the sintering process. The diffraction pattern of the bonding phase shows FCC (face centered cubic) structure, i.e., austenite, which is the same as the expected result. No other structural compounds were found in the bonding phase. The bonding interface between the binder and TiC particles is close, which shows that it has a good metallurgical bonding.

In this study, the content of Mn element is in the range of composition that makes Fe form a single austenite-like high manganese steel, while the content of Mo, Cr, Ni and other alloy elements is in the range of solid solution with the Fe element. Therefore, during the process of sintering or heat treatment, all alloy elements are completely in a solid solution in the bonding phase of the austenite structure. The single austenite phase has a high toughness while the solid solution strengthening of Mn, Mo, Cr, Ni gives the bonding phase high strength and hardness. Thus, the TRS and IM improvement of the alloy are attributed to the uniform distribution of the alloying elements, refined grain structure. In particular, the IM improvement of the alloy is mainly attributed to the inhomogeneous structure of the alloy. Therefore, this is an instructive attempt to produce high impact toughness TiC-high Mn steel-bonded carbide.

## 5. Conclusions

A cellular TiC-high Mn steel bonded carbide was designed and fabricated using powder metallurgy techniques. The main conclusions are as follows:(1)A cellular structure consisting of fine-grained TiC zones distributed in the coarse-grained TiC ceramic matrix was obtained.(2)The binder of the alloy is FCC (face centered cubic) high-Mn steel with good toughness, though Fe-Mo and Fe-Mo-Cr-C pre-alloyed powders were used as the binder of coarse-grained TiC zones and fine-grained TiC zones, respectively.(3)When the starting ratio of MP-A to MP-B was 60:40, the alloy reached the maximum TRS and IM at 2231 MPa and 12.87 J/cm^2^, respectively. Thus, this paper provides a template for preparing high-strength and high-toughness TiC-high steel-bonded carbide.

## Figures and Tables

**Figure 1 materials-13-00757-f001:**
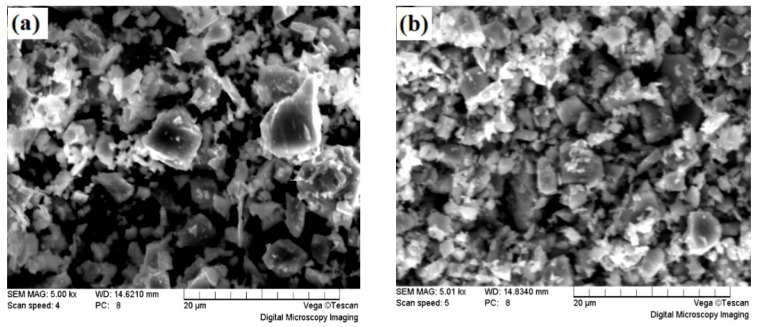
SEM morphology of TiC particles, (**a**) coarse TiC, (**b**) fine TiC.

**Figure 2 materials-13-00757-f002:**
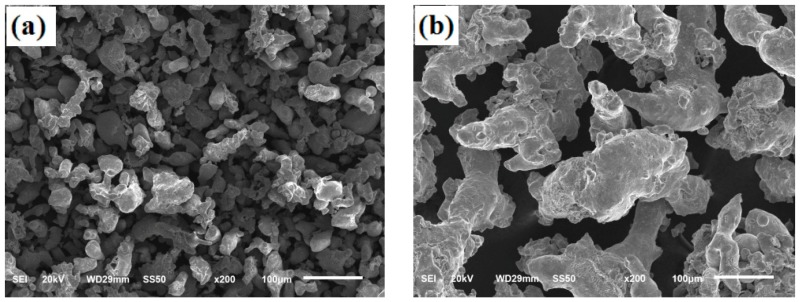
SEM morphology of pre-alloyed iron powders, (**a**) Fe-1.5Mo, (**b**) Fe-3.0Mo-3.75Cr-0.7C.

**Figure 3 materials-13-00757-f003:**
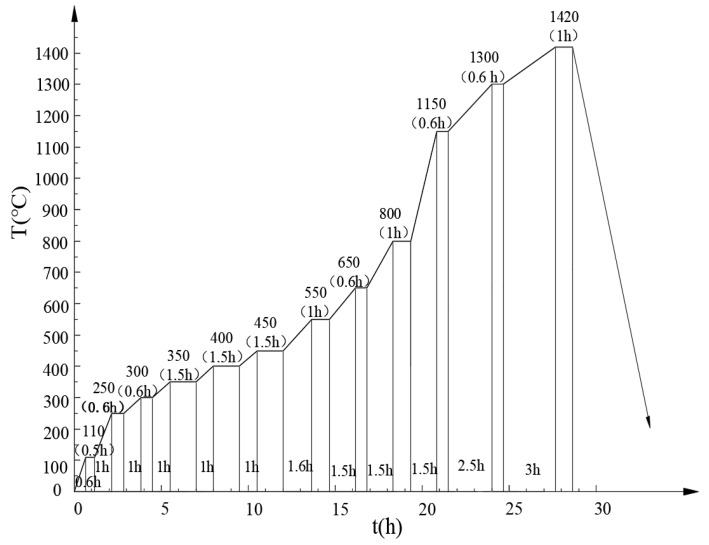
The integrated dewaxing and sintering curve of the alloy.

**Figure 4 materials-13-00757-f004:**
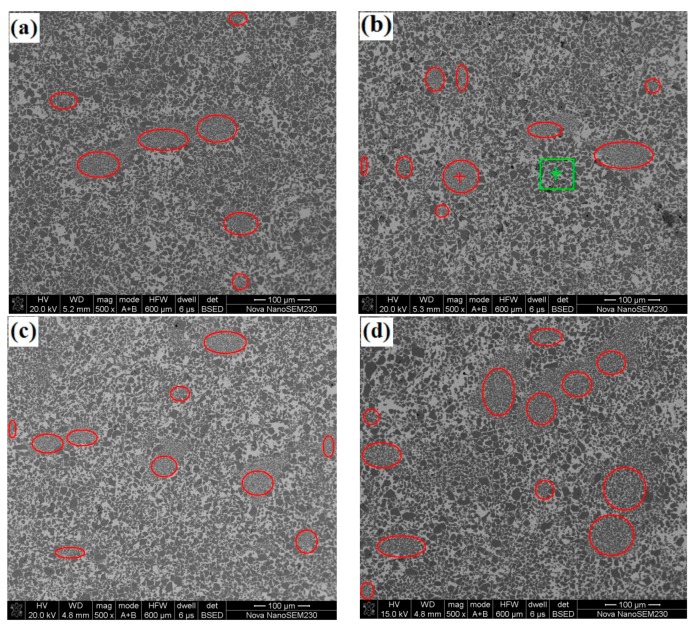
SEM images of the experimental samples, (**a**) #1, (**b**) #2, (**c**) #3, and (**d**) #4.

**Figure 5 materials-13-00757-f005:**
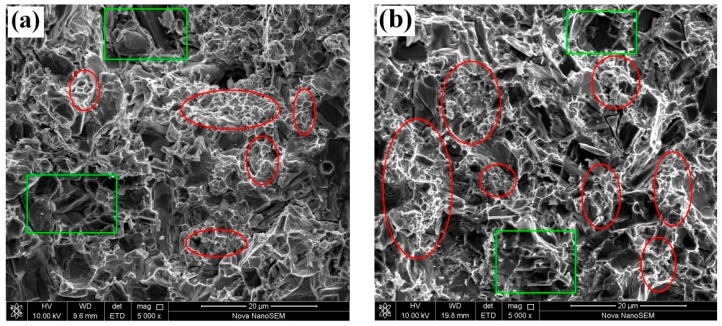
Impact fracture surface morphology of (**a**) Sample #1 and (**b**) Sample #2.

**Figure 6 materials-13-00757-f006:**
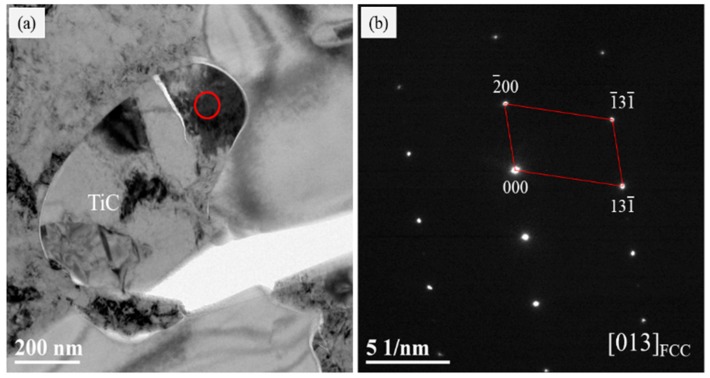
TEM electron diffraction pattern analysis of Sample #2, (**a**) TEM image of coarse-grained TiC zone, (**b**) Electron diffraction pattern of binder marked by red circle in Figure 6a; (**c**) TEM image of fine-grained TiC zone, (**d**) Electron diffraction pattern of binder marked by the red circle in Figure 6c.

**Table 1 materials-13-00757-t001:** Main characteristics of the raw powders in the study.

Powders	Particle Size (μm)	Purity (wt.%)	Oxygen (wt.%)	Manufacturer
Coarse TiC	3.1–3.3 ^①^	≥99.5	≤0.25	Zhuzhou Cemented Carbide Group Co., Ltd., China
Fine TiC	0.8–1.5 ^①^	≥99.4	≤0.28	
Ni	43	≥99.8	<0.12	Shanghai Xtnami Science & Technology Co., Ltd., China
FeMn81.4	74	≥99.5 ^②^	<0.15	Jinzhou Honda New Material Co., Ltd., China
Graphite	30	≥99.8	<0.05	Qingdao Baichuan Graphite Co., Ltd., China
Fe-1.5Mo	74	≥99.5 ^③^	<0.28	Laiwu Iron and Steel Group Powder, Metallurgy Co., Ltd., China
Fe-4.5Mo-3.75Cr-0.7C	147	≥99.0 ^④^	<0.30	Own Manufacturing

① Fisher particle size; ② The total components of Fe and Mn; ③ The total components of Fe and Mo; ④ The total components of Fe, Mo, Cr and C.

**Table 2 materials-13-00757-t002:** Composition and proportions of ingredients of MP-A and MP-B (wt.%).

Mixed Powder	TiC	Ni	Mo	Mn	C	Pre-Alloyed Iron Powder
MP-A	50	2.25	0.6	7.0	0.7	Bal. (Fe-1.5Mo)
MP-B	46	2.25	-	7.0	0.41	Bal. (Fe-4.5Mo-3.75Cr-0.7C)

**Table 3 materials-13-00757-t003:** Relationship of experimental samples using MP-A and MP-B (wt.%).

Sample	#1	#2	#3	#4
MP-A	80	60	40	20
MP-B	20	40	60	80

**Table 4 materials-13-00757-t004:** Preparation processes of the cellular TiC-high Mn steel-bonded carbide.

Procedure	Process Name	Main Contents and Parameters
1	Weighing and mixing	Weighing the ingredients of MP-A and MP-B, and mixing for 120 min.
2	Wet-milling	Ball to powder weight ratio: 3:1 for MP-A, 6:1 for MP-B; milling time: 24 h; protective medium in milling: ethanol
3	Pre-granulation	Adding 4 wt.% rubber; block granulation method; 60 mesh
4	Weighing and mixing	Weighing MP-A and MP-B; remixing for 120 min
5	Re-granulation	Adding 2 wt.% rubber; block granulation method; 20 mesh
6	Pressing of green compacts	200 MPa uniaxial pressing; Φ20 × 60 mm^3^
7	Sintering	Sintering temperature: 1420 °C for 60 min

**Table 5 materials-13-00757-t005:** EDS analysis result of the fine-grained TiC zone and coarse-grained TiC zone.

Elements Content, wt.%	C	Mo	Ti	Cr	Mn	Fe	Ni	Total
Coarse-grained TiC zone	10.47	0.85	46.64	0.26	5.13	32.72	3.92	100
Fine-grained TiC zone	10.27	2.69	42.25	5.90	4.89	30.79	3.22	100

**Table 6 materials-13-00757-t006:** Relative density of thecellular TiC-high Mn steel-bonded carbides.

Samples	#1	#2	#3	#4
Relative density (%)	98.57	98.51	98.48	98.33

**Table 7 materials-13-00757-t007:** Properties of the cellular TiC-high Mn steel-bonded carbides.

Sample	Hardness (HRC)	TRS (MPa)	IM (J/cm^2^)
#1	63.2	2078	9.17
#2	64.3	2231	12.87
#3	64.1	2206	12.11
#4	62.7	2027	9.05

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
