# Peer review of "The Preparation Process, Microstructure and Properties of Cellular TiC-High Mn Steel-Bonded Carbide"

_materials, 2020, doi:10.3390/ma13030757_

Round 1
Reviewer 1 Report
The state-of-the-art and references must to be improved since no references for the years 2018 and 2019.
Why did mention the honeycomb structure? The structure seems to be a cellular one … and if is true you must to evidence it using high magnitude by SEM. Figure 4 is unclear and is not convincing!!
Line 314: explain TRS and IM mechanism; what is about?!
Which can be destination in exploitation for this kind of material?!
Author Response
Thank you for the comments very much. my reply is in the attached file.

Reviewer 2 Report
The paper presents a description of the proposed manufacturing technology for TiC–high Mn steel-bonded carbide.
The paper is consistent and the presented illustrative material is well elaborated.
The authors refer to 30 literature items describing the issue. The referred items are appropriate and well selected.
In the introduction, the authors describe the use of modern sintered materials – lines 24-41.
In practice, the use of sintered materials requires dimensional and shape accuracy of the manufactured parts. I suggest that the paper relate to this problem.
In line 148, the authors describe the preparation of the obtained specimens for measurement. Please describe how the specimens were machined.
In lines 114 and 195, I have noticed words written in a different typeface.
Author Response

(The authors gave the same response as above.)

Round 2
Reviewer 1 Report
After ENglish spelling and grammar it can be published!